# The role of results in deciding to publish: A direct comparison across authors, reviewers, and editors based on an online survey

**Jasmine Muradchanian** \*, **Rink Hoekstra, Henk Kiers, Don van Ravenzwaaij**

Behavioural and Social Sciences, University of Groningen, Groningen, the Netherlands

* jasmine.muradchanian@gmail.com

## Abstract

### Background

Publishing study results in scientific journals has been the standard way of disseminating science. However, getting results published may depend on their statistical significance. The consequence of this is that the representation of scientific knowledge might be biased. This type of bias has been called publication bias. The main objective of the present study is to get more insight into publication bias by examining it at the author, reviewer, and editor level. Additionally, we make a direct comparison between publication bias induced by authors, by reviewers, and by editors. We approached our participants by e-mail, asking them to fill out an online survey.

### Results

Our findings suggest that statistically significant findings have a higher likelihood to be published than statistically non-significant findings, because (1) authors ($n = 65$) are more likely to write up and submit articles with significant results compared to articles with non-significant results (median effect size 1.10, $BF_{10} = 1.09*10^7$); (2) reviewers ($n = 60$) give more favourable reviews to articles with significant results compared to articles with non-significant results (median effect size 0.58, $BF_{10} = 4.73*10^2$); and (3) editors ($n = 171$) are more likely to accept for publication articles with significant results compared to articles with non-significant results (median effect size, 0.94, $BF_{10} = 7.63*10^7$). Evidence on differences in the relative contributions to publication bias by authors, reviewers, and editors is ambiguous (editors vs reviewers: $BF_{10} = 0.31$, reviewers vs authors: $BF_{10} = 3.11$, and editors vs authors: $BF_{10} = 0.42$).

### Discussion

One of the main limitations was that rather than investigating publication bias directly, we studied potential for publication bias. Another limitation was the low response rate to the survey.

**Data Availability Statement:** Data and R codes are available within the OSF repository: https://osf.io/4c6ev/. Additionally, they have been uploaded as part of supplementary material.

**Funding:** The author(s) received no specific funding for this work.

**Competing interests:** The authors have declared that no competing interests exist.

## Introduction

The standard dissemination method of study results is their publication in scientific journals. However, not all study results are being published, meaning that our published literature is a selective sample of scientific knowledge as a whole. When publishability of scientific findings is related to the quality of the study, this will typically be considered acceptable, or even desirable. However, it may also relate to characteristics that are not necessarily related to the quality of the study, such as the statistical significance of the findings, or whether the findings support widely accepted theories. If this forms the basis for differential publication, then this will result in a selection bias in the sense that the published scientific knowledge is not a good representation of the scientific knowledge that actually exists [1]. This type of bias has been called publication bias [e.g., 1–4].

The consequence of publication bias in scientific literature is that it distorts the assessment of knowledge regarding a certain phenomenon. More specifically, studies supporting the prevailing 'Zeitgeist' might be overrepresented, and the magnitude of a relation or effect might be overestimated [2, 3, 5, 6]. In addition, publication bias has been perceived as a threat to the validity of synthesis methodologies such as meta-analysis [e.g., 3, 5, 7, 8 p. 436]. Various statistical methods have been developed for detecting the presence of publication bias, and for adjusting the meta-analytic estimate for the presence of publication bias in the sample [9]. In a recent study by Sladekova, Webb, and Field [9], it was evaluated how applying appropriate publication bias adjustment methods changed the effect size estimates of published meta-analyses. The authors reanalysed datasets from published articles in psychology journals, and they concluded that the adjustment found for the different techniques was small. However, they emphasized that this does not mean that the levels of publication bias are low in psychology.

Publication bias has been reported in a variety of disciplines within social sciences (including economics [1, 3, 4, 6, 10]) and medical sciences [1, 7, 11, 12]. The awareness of the existence of this type of selection bias in published literature goes back to at least 1959, when Sterling stated that studies reporting statistically significant findings have a higher probability of being published than studies reporting statistically non-significant findings [13]. Since then, many authors have been focusing on studying publication bias (see, for example, Dwan et al. [12] for a systematic review on publication bias, and Song et al. [14] for a review of publication and related biases), and in particular the finding of statistically significant outcomes having a higher likelihood of getting published than null results has been confirmed in subsequent studies [e.g., 6, 7, 11]. Additionally, there is evidence that among studies that do get published, those that are reporting statistically significant results are published more quickly than those that are reporting statistically non-significant results [e.g., 7]. Interestingly, in a recent study, conducted by Turner et al. [15] about reporting bias (that also includes publication bias) in anti-depressants literature, it appeared that although reporting bias seems to persist, it has diminished for newer anti-depressants compared to older ones.

Although prejudice against null results seems to be an important filter in publication bias, there are also other biasing filters [2, 5], which makes the entire publication bias a complex phenomenon that is difficult to define. For example, Cooper, DeNeve, and Charlton [2] found that a great number of statistically significant results was not submitted for presentation and publication by researchers in their sample. Another potential filter for publishing studies is the direction of study results [16]. However, in a publication, such mechanisms that cause bias typically cannot be accurately specified and quantified, so it is difficult to obtain appropriate adjustments that can be used, for example, in meta-analytic datasets. In addition, the mechanisms might differ across datasets and subject areas [8 p. 437]. As statistical significance of study results forms a major source of publication bias, we decided to focus on this aspect in the present study.

So far, it seems that some direct evidence for publication bias (i.e., directly comparing published and unpublished scientific literature, or following a cohort of scientific studies from their inception [17]) has been provided at both journal (i.e., reviewer and editor) and author level. Additionally, some research has specifically focused on publication bias induced by authors, reviewers, or editors separately [6, 11, 18–20]. Regardless of who has been held responsible for publication bias, it is important to note that it does not occur from a deliberate motive to deceive [3], and academics might not even be aware of these biases [21]. Some have suggested that publication bias may be more pernicious at the author level than at the journal level [10], and some have specifically mentioned that editorial decisions contribute to publication bias [3, 17]. However, to the best of our knowledge, this has not been studied directly yet. In the present study, we would like to focus on authors, reviewers, and editors simultaneously in order to see where exactly and to what extent in the process of generating a scientific paper publication bias occurs. Additionally, we would like to make a direct comparison between publication bias induced by authors, reviewers, and editors. By doing so, we will make an attempt to answer the following research question: To what extent do authors, reviewers, and editors contribute to publication bias in scientific literature? We will explicitly focus on whether statistically significant findings have a greater likelihood of being published than statistically non-significant findings.

## Method

### Ethics statement

This study has been approved by the ethics committee of the University of Groningen (approval number: PED-2122-S-0074) and it complies with all relevant ethical regulations of the University of Groningen. Written informed consent was obtained from all participants.

### Participants

Our sampling plan, stimulus list and materials, and planned analyses were preregistered before data collection (see https://osf.io/ug2nk). In the present study, the target population consisted of editors, reviewers, and authors from various scientific disciplines. The editors in our prospective sample were a selection of editors representing 1500 journals who were already approached in a study by Hamilton et al. [22]. Hamilton and colleagues targeted the following fields: ecology, economics, medicine, physics, and psychology, and they excluded books, data repositories, and non-English journals. Then the authors selected 300 journals within each field with the highest impact factor. For the present study, we excluded editors from physics ($n$ = 300) because we believe that significance testing plays a different role in research in this field compared to the fields we did include. Furthermore, we excluded four duplicates, resulting in a final list with 1196 editors, who were invited to fill out our questionnaire. The editors were recruited to the study between 13 July 2022 and 28 September 2022.

Per editor in our sample (i.e., 210 editors who at least partly filled out our survey in Qualtrics), we randomly selected six authors who published in the journal during the time this editor was editing this journal. In total, after eliminating duplicates, we contacted 1196 editors and 1215 authors/reviewers. By doing this, there was a link between the editors and authors/reviewers in our sample, meaning that they have knowledge about the same field. We contacted them by e-mail. In the e-mail we mentioned that we were looking for academics who have reviewed other researchers' work at least once in their academic career. Then, half of these authors were asked to fill out our survey from an author perspective. The other half of these authors were asked to fill out our survey from a reviewer perspective. The assignment

was done randomly. The authors/reviewers were recruited to the study between 29 September 2022 and 30 October 2022.

In our analyses, we only included the answers of the respondents who answered the question about the significant result and the question about the non-significant result (see below in the subsection "Materials and procedure" for detailed information about the questions included in our surveys). We collected our samples in a stepwise way: 1) We invited 1196 editors, of which 171 responded to the survey. 2) We invited 1215 researchers for the role of reviewer or author, of which 125 responded. 3) Qualtrics randomly assigned each of the respondents in step 2 to the roles of Author (65) and Reviewer (60). The eventual sample sizes for authors, reviewers, and editors were 65, 60, and 171 respectively. The final response rate was 10.3% for the authors/reviewers and 14.3% for the editors.

## Materials and procedure

The editors and the authors/reviewers included in our list received an email with a Qualtrics survey link. The survey was anonymous, meaning that the respondents could not be identified based on their answers.

The survey started with asking informed consent for all respondents. Then they got presented with a hypothetical scenario in which two studies were described, one with a statistically significant result ($p = .02$) and one with a statistically non-significant result ($p = .26$). We asked the respondents questions about their intention to write up and submit a paper (author level), recommend publication of a paper (reviewer level), and accept a paper for publication (editor level). Exact formulations of each question are displayed in S1 Table. They were able to provide their answers by using a slider on a scale ranging from 0% to 100%. Approximately half of the respondents received the question about the significant result first, and the other half received the question about the non-significant result first.

After completing this part of the survey, respondents received the following optional (qualitative) question, with a blank space where they could type in their answer:

> *"Please explain briefly why you gave these answers to the previous questions."*

Finally, we were interested in how academics deal with significance testing. Therefore, we included the following question to our survey, and the respondents indicated their answer by using a slider on a scale ranging from 1 to 10:

> *"Please indicate how familiar you are with significance testing on a scale ranging from 1 (I have never heard about significance testing before) to 10 (I am able to interpret significance test outcomes correctly)."*

## Analyses

Our analysis plan was preregistered (see https://osf.io/ug2nk); we deviated from our planned analysis in two respects and will describe this further down as we describe our actual analyses. To answer our research question ("To what extent do authors, reviewers, and editors contribute to publication bias in scientific literature?"), we first computed a difference score (in percentage points) in the likelihood of endorsing publication between the significant and the non-significant results for all respondents (from here on DS).

**Bayesian one sample *t* test.** First, we performed a Bayesian one sample *t* test to determine for the three scientific roles separately the difference in the likelihood of endorsing publication

between the significant and the non-significant results. The null hypotheses for the various *t* tests stated that there is no difference in the likelihood of endorsing publication between the significant and the non-significant results, and the alternative hypotheses stated that there is a difference in the likelihood of endorsing publication between the significant and the non-significant results.

**Bayesian independent samples *t* test.** Second, we performed a Bayesian independent samples *t* test in order to determine the differences in the average DS, computed initially, across scientific roles (i.e., editors vs reviewers, reviewers vs authors, and editors vs authors). The null hypotheses for the various *t* tests stated that there is no difference in the average DS between the two scientific roles, and the alternative hypotheses stated that there is a difference in the average DS between the two scientific roles.

For each test we conducted, we calculated Bayes factors that quantify the relative evidence for the alternative hypothesis over the null hypothesis ($BF_{10}$) provided by the data, and 95% credible intervals based on posterior distributions of the standardized effect size parameters. We decided to conduct Bayesian *t* tests, as they allow for quantification of evidence in favour of the null hypothesis (relative to a composite alternative hypothesis). The prior distribution for the standardized effect size under the alternative hypothesis was a Cauchy distribution centred on zero with scale parameter $1/\sqrt{2}$, which is the default prior that is often used in Bayesian null hypothesis testing, and thus makes results comparable across different studies. Both the Bayesian one sample *t* tests and the Bayesian independents samples *t* tests were performed in JASP (version 0.16.1 [23]).

It turned out that the DS in the likelihood of endorsing publication between significant and non-significant results were extremely non-normally distributed within each of the groups of editors, reviewers, and authors. As such, we deviated from our preregistered analysis plan by additionally conducting a non-parametric Bayesian alternative, that relaxes the assumption of normally distributed data, to both the one sample *t* tests (the Bayesian signed-rank test [24]) and the independent samples *t* tests (the Bayesian rank-sum test [24]) mentioned above. These tests were performed in RStudio (version 4.1.3 [25]), using the R scripts by van Doorn et al. ([24] see https://osf.io/gny35/). For the prior specification, a Cauchy distribution centred around zero having a width parameter of $1/\sqrt{2}$ was used. The Bayes factors and the 95% credible intervals can be interpreted in the same way as in the corresponding parametric approach.

**Qualitative question.** Furthermore, we explored the answers of the respondents to the question on why they gave the answers to the questions about the significant and non-significant results in order to see whether there were specific patterns in the respondents' reasoning. Based on the answers provided by the respondents to the qualitative question, we made five categories: 1 = *p*-value is decisive, 2 = *p*-value is important/relevant but not decisive, 3 = *p*-value is not important/relevant, 4 = other/unclear (rest category), and 0 = no answer provided to the qualitative question. Authors JM and RH independently classified the answers to the qualitative question into one of these categories. For 45 out of (a total of) 296 answers, there was disagreement on the classification. For each answer for which there was disagreement on classification, authors JM and RH reached consensus in a discussion of no longer than 5 minutes.

**Familiarity with significance testing.** Finally, in our preregistered analysis plan, we were planning to explore to what extent the difference in likelihood of accepting a significant versus a non-significant result was related to familiarity with significance testing. The scores on the familiarity with significance testing question appeared to have a ceiling effect. Therefore, we decided to drop this analysis.

## Results

To study to what extent authors, reviewers, and editors contribute to publication bias in scientific literature, we computed the difference in the reported likelihoods to endorse publication between the significant and the non-significant results for all respondents. The distributions of these differences for the three scientific roles are provided in Fig 1. It is clear that all three distributions are strongly right skewed.

We interpreted differences between -5 and +5 as practically no difference in the reported likelihoods to endorse publication. Following this definition, we found that 45% of the authors (29 out of 65), 60% of reviewers (36 out of 60), and 48% of editors (82 out of 171) showed practically no difference in the reported likelihoods to endorse publication between the significant and the non-significant results. As is clear from Fig 1, the other half of the respondents

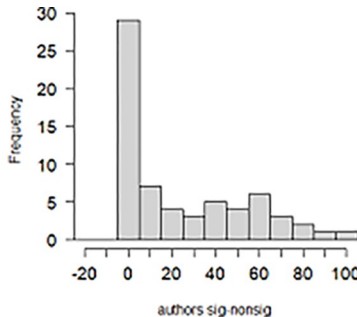

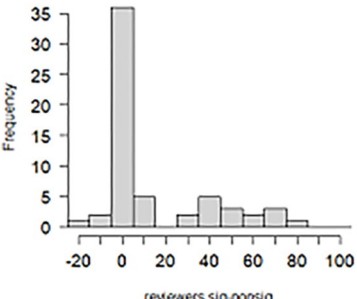

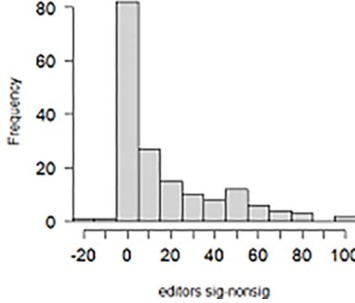

**Fig 1. The distribution of the difference in the reported likelihoods to endorse publication between the significant and the non-significant results for authors (panel 1), reviewers (panel 2), and editors (panel 3).**

(*N* = 144) indicated that it *does* matter, and that papers with significant results are more likely to be endorsed for publication than papers with non-significant results. A peculiar finding was that there were five respondents with negative differences more extreme than -5: -10 (twice), -11, and -20 (twice). These respondents thus indicated that they would be more likely to endorse publication of non-significant results than of significant results. To the optional qualitative question about why these respondents gave these answers, one respondent indicated that they can get suspicious of questionable research practices when the *p*-value is just below .05. Based on the answer of another person, it became clear that the negative DS was a representation of practically no difference in the reported likelihoods to endorse publication. The other three participants did not provide an answer to the qualitative question, so we did not know why they would be more likely to endorse publication of non-significant results than of significant results.

## Bayesian one sample *t* test

We performed a Bayesian one sample *t* test to determine for the three scientific roles separately whether the significance of the results affected the likelihood of endorsing publication. The results are presented in Table 1 (see column "Parametric"); the posterior distributions for the standardized effect sizes of differences between significant and non-significant results for the three scientific roles, given the used priors, can be found in S1–S3 Figs. Based on the Bayes factors and the credible intervals reported in Table 1 (column "Parametric"), it can be concluded there is extreme evidence that the average DS in the likelihood of endorsing publication between significant and non-significant results differed at the population level for the three scientific roles. In other words, statistically significant findings seemed to have a higher likelihood to be published than statistically non-significant findings.

The results of the Bayesian signed-rank test [24] are presented in Table 1 (see column "Non-parametric"). The conclusions based on the Bayesian signed-rank test were the same as the conclusions based on the parametric approach: extreme evidence for the hypothesis that statistically significant findings have a higher likelihood to be published than statistically non-significant findings. The effect sizes were large for all three scientific roles.

## Bayesian independent samples *t* test

To examine whether there were differences in the beneficial treatment of significant over non-significant results between editors, reviewers, and authors (i.e., editors vs reviewers, reviewers vs authors, and editors vs authors), we performed three Bayesian independent samples *t* tests. The results are presented in Table 2 (see column "Parametric"); the posterior distributions for the standardized differences of differences, given the used priors, can be found in S4–S6 Figs. In Table 2 (column "Parametric"), we see that for the group of editors versus reviewers, the

**Table 1. Parametric and non-parametric results for the differences in the likelihood of endorsing publication between the significant and the non-significant results for the three scientific roles separately.**

| Scientific role | n | Mean | SE | Parametric | | | | Non-parametric | | | |
|---|---|---|---|---|---|---|---|---|---|---|---|
| | | | | | | 95% credible interval | | | | 95% credible interval | |
| | | | | $BF_{10}$ | Effect size | Lower | Upper | $BF_{10}$ | Effect size | Lower | Upper |
| Authors | 65 | 24.99 | 3.60 | $4.76*10^6$ | 0.84 | 0.55 | 1.12 | $1.09*10^7$ | 1.10 | 0.72 | 1.44 |
| Reviewers | 60 | 14.33 | 3.23 | $5.08*10^2$ | 0.55 | 0.28 | 0.82 | $4.73*10^2$ | 0.58 | 0.30 | 0.87 |
| Editors | 171 | 17.25 | 1.81 | $3.51*10^{14}$ | 0.72 | 0.55 | 0.89 | $7.63*10^7$ | 0.94 | 0.68 | 1.15 |

*Note*: Effect size is the median value for the standardized effect sizes (Cohen's d).

**Table 2. Parametric and non-parametric results for the differences in the average DS across scientific roles.**

| Scientific roles | Parametric | | | | Non-parametric | | | |
|---|---|---|---|---|---|---|---|---|
| | | | 95% credible interval | | | | 95% credible interval | |
| | $BF_{10}$ | Effect size | Lower | Upper | $BF_{10}$ | Effect size | Lower | Upper |
| Editors vs reviewers | 0.22 | -0.11 | -0.40 | 0.17 | 0.31 | -0.17 | -0.46 | 0.12 |
| Reviewers vs authors | 1.64 | 0.36 | 0.02 | 0.71 | 3.11 | 0.42 | 0.07 | 0.78 |
| Editors vs authors | 1.24 | 0.29 | 0.01 | 0.57 | 0.42 | 0.19 | -0.08 | 0.47 |

*Note*: Effect size is the median value for the standardized effect sizes (Cohen's d).

Bayes factor suggested that the null hypothesis was a little over four times more likely than the alternative hypothesis in light of the data (i.e., $1/0.22 \approx 4.5$). However, the associated credible interval is quite wide, indicating that there is much uncertainty about this difference and its direction.

Regarding the difference in the average DS across reviewers versus authors and editors versus authors, although the alternative hypothesis was slightly favoured over the null hypothesis, the data did not provide sufficiently strong evidence to reject or accept either hypothesis. In other words, the present findings do not give sufficiently clear indications whether or not at population level there are differences in average DS across the three scientific roles.

The results of the Bayesian rank-sum test [24] are presented in Table 2 (see column "Non-parametric"). The conclusions based on the Bayesian rank-sum test were the same as the conclusions based on the parametric approach: the present findings do not give sufficiently clear indications whether or not at population level there are differences in average DS across the three scientific roles.

## Qualitative question

Of respondents who provided a response on the importance of *p*-values, 36.1% found them decisive or important for publication, whereas 37.8% did not (see Table 3 column 2).

For each category of the qualitative question, we computed the average DS in the likelihood of endorsing publication between the significant and the non-significant results (see Table 3 column 3). We performed exploratory analyses in which we conducted Bayesian independent samples *t* tests in JASP in order to test the difference in the average DS between category 1 (*p*-value is decisive) versus category 2 (*p*-value is important/relevant but not decisive), and between category 2 versus category 3 (*p*-value is not important/relevant). The null hypotheses for the two *t* tests stated that there is no difference in the average DS between the two categories, and the alternative hypotheses stated that there is a difference in the average DS between the two categories. For each test we conducted, we calculated Bayes factors that quantify the

**Table 3. Classification answers to the qualitative question.** Column three contains the average difference in likelihood of endorsing publication of an article with significant versus non-significant results per subgroup of participants categorized based on their qualitative answers.

| Category | N | Average difference | 95% confidence interval for average difference |
|---|---|---|---|
| 1 (*p*-value is decisive) | 35 (11.8%) | 51.87 | [43.40, 60.34] |
| 2 (*p*-value is important/relevant but not decisive) | 72 (24.3%) | 18.38 | [13.76, 22.99] |
| 3 (*p*-value is not important/relevant) | 112 (37.8%) | 4.66 | [2.33, 6.98] |
| 4 (other/unclear [rest category]) | 43 (14.5%) | 22.81 | [13.89, 31.74] |
| 0 (no answer provided to the qualitative question) | 34 (11.5%) | 23.31 | [13.25, 33.37] |
| Total | 296 (100%) | 18.36 | [15.45, 21.26] |

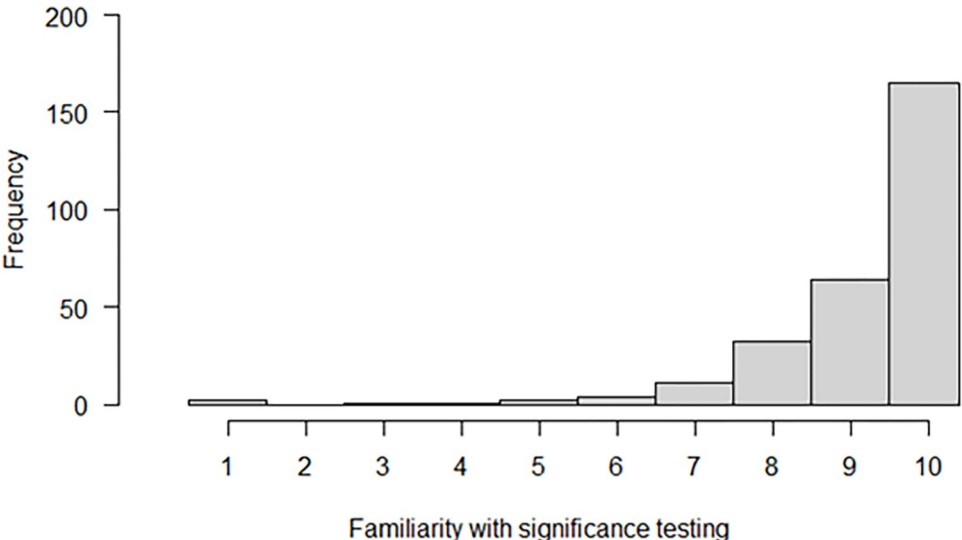

**Fig 2. Histogram for the responses on the question about familiarity with significance testing.**

relative evidence for the alternative hypothesis over the null hypothesis ($BF_{10}$) provided by the data, and 95% credible intervals based on posterior distributions of the standardized effect size parameters. The results showed that the higher the importance of the *p*-value indicated in the answer to the qualitative question, the larger the average DS in the likelihood of endorsing publication between the significant and the non-significant results (category 1 versus category 2: $BF_{10}$ = 5.09*$10^8$, 95% credible interval = 1.05, 1.97; category 2 versus category 3: $BF_{10}$ = 4.02*$10^5$, 95% credible interval = 0.54, 1.16). This means that the answers to the qualitative question were in line with the answers to the quantitative questions about the significant and non-significant results. Furthermore, it appeared that the average DS in the categories 4 (i.e., other/unclear [rest category]) and 0 (i.e., no answer provided to the qualitative question) was quite large too.

## Familiarity with significance testing

As mentioned at the end of the method section, we were planning in our preregistration to explore to what extent the difference in likelihood of accepting a significant versus a non-significant result was related to familiarity with significance testing. We decided to drop this analysis because, as can be observed in Fig 2, the scores on the familiarity with significance testing question appeared to have a ceiling effect.

## Discussion

In the present study, we studied publication bias by examining it at the author, reviewer, and editor level simultaneously. We asked a sample of academics in their role as either editor, reviewer, or author to fill out a survey in order to see whether they were more likely to accept for publication (editor), recommend publication of (reviewer), or write up/submit (author) an article with significant versus non-significant statistical results. We also made a direct comparison between these potential differences across the three scientific roles. We think that the advantage of getting insights into publication bias simultaneously is that it enhances the comparability between publication bias induced by authors, by reviewers, and by editors. By doing it simultaneously, it was possible to keep the fictitious scenarios constant across the three roles,

and in turn, it was possible to make a direct comparison between publication bias induced by the three scientific roles.

We learned from the obtained results that academics indicated they were more likely to endorse an article with statistically significant findings for publication than an article with statistically non-significant findings; this difference was fairly large, and it was the case for all three scientific roles separately. However, we did not get a clear indication from the present findings whether or not, at population level, these differences differed across the three scientific roles.

There were several limitations in the present study. Firstly, we focused on one ingredient of publication bias, namely the role of statistical significance of the results. Although this seems to form a major source of publication bias, there may well be other factors that contribute. For example, in addition to the statistical significance, academics might be biased against studies reporting findings that are not in line with their theoretical views [16]. Another mechanism, which is related to language, is that academics from countries where English is not the first language might publish their null findings in national journals that are not in English. For example, Egger et al. [26] found that randomized control trials were more likely to be published in an English-language journal if their findings were statistically significant, whereas other trials were reported in local journals in German. As a consequence, English-language bias might be introduced in meta-analyses and reviews if only English-language published trials are included.

There were some limitations related to the surveys used in the present study. The most important one is that some participants mentioned, for instance, via email, that insufficient amount of information (e.g., about the study design, etc.) was provided in the scenarios, or that some relevant information was missing. We had decided to leave out contextual information such that the respondents could only focus on the *p*-value (assuming that both studies had, for example, a high quality). Additionally, because of this, the purpose of our study was fairly obvious (some respondents let us know by email that they were aware that we were studying publication bias). Perhaps the transparency of the study aim was related to the next limitation of our study, namely the low response rate. We think that there is a possibility that academics might have experienced our survey and the topic of publication bias as uncomfortable, which in turn, resulted in a low response rate. Furthermore, we evaluated only two *p*-values, namely $p = .02$ and $p = .26$. It would have been interesting to also examine more minor deviations such as $p = .049$ and $p = .051$ in order to see what the differences are in the likelihood of endorsing the statistically significant finding for publication compared to the statistically non-significant finding.

Social desirability might have played a role in the present context. It is possible that some academics systematically refrain from publishing non-significant findings without acknowledging this practice. If this is the case, then our results likely underestimate the reality.

Finally, in the present study, we did not investigate publication bias directly. We studied potential for publication bias based on questionnaires in which we asked what academics would do after reading a simplified fictitious situation. It is important to note that respondents' answers do not necessarily represent actual behaviour.

Allowing for these limitations, our findings seem to suggest that authors, reviewers, and editors all contribute to publication bias, and that there does not seem to be a single party who is responsible for producing a scientific literature that is biased. This does make sense because there is an overlap between authors, reviewers, and editors. In other words, these groups are not mutually exclusive. For example, authors are often reviewers, and can be editors as well. Furthermore, it is possible that editors follow the suggestions of reviewers, that authors behave according to the expectations they have about the opinions of reviewers and editors, and that

reviewers give advice based on advice they think they would receive themselves. In other words, the three groups are clearly dependent of one another, so we shouldn't assume an independent contribution to publication bias. Moreover, it might well be that the different actors in this interplay already change their behaviour based on anticipations of what subsequent actors expect them to do. Authors, for example, could pre-emptively remove certain non-significant analyses because they expect future reviewers to criticize those, and reviewers may suggest selective publishing of certain significant findings anticipating that editors expect them to. Hence, we think we should be careful in drawing premature conclusions about who's responsible for publication bias based on our results.

One of the proposed solutions in the literature to combat publication bias is employing Registered Reports, and the results found in the present study highlight the importance Registered Reports even more. The idea behind Registered Reports is that a significant part of an eventual paper is reviewed before data collection. Registered Reports are thought to be immune to publication bias. This is because the decision whether to accept or reject the manuscript is based on the importance of the research question and the quality of the methods; it is never based on the statistical significance of the results [27].

As already suggested by Franco, Malhotra, and Simonovits [6], our results show that an important part of establishing solutions on an institutional level in order to improve scientific openness and transparency would be a deeper understanding of motivations of academics who decide to publish articles as a function of results. Perhaps future research can focus on what authors, reviewers, and editors need in order to let publication decisions depend on the practical significance of the research questions, instead on statistical significance of their outcomes.

## Supporting information

**S1 Table. The scenarios and questions across the three scientific roles.**
(DOCX)

**S1 Fig. Posterior distribution for the standardized effect size of difference between significant and non-significant results for authors, given the used priors.**
(TIF)

**S2 Fig. Posterior distribution for the standardized effect size of difference between significant and non-significant results for reviewers, given the used priors.**
(TIF)

**S3 Fig. Posterior distribution for the standardized effect size of difference between significant and non-significant results for editors, given the used priors.**
(TIF)

**S4 Fig. Posterior distribution for the standardized differences of differences across editors versus reviewers, given the used priors.**
(TIF)

**S5 Fig. Posterior distribution for the standardized differences of differences across reviewers versus authors, given the used priors.**
(TIF)

**S6 Fig. Posterior distribution for the standardized differences of differences across editors versus authors, given the used priors.**
(TIF)

## Author Contributions

**Conceptualization:** Jasmine Muradchanian, Rink Hoekstra, Henk Kiers, Don van Ravenzwaaij.

**Formal analysis:** Jasmine Muradchanian, Rink Hoekstra, Henk Kiers, Don van Ravenzwaaij.

**Investigation:** Jasmine Muradchanian.

**Methodology:** Jasmine Muradchanian, Rink Hoekstra, Henk Kiers, Don van Ravenzwaaij.

**Project administration:** Jasmine Muradchanian.

**Supervision:** Rink Hoekstra, Henk Kiers, Don van Ravenzwaaij.

**Writing – original draft:** Jasmine Muradchanian.

**Writing – review & editing:** Jasmine Muradchanian, Rink Hoekstra, Henk Kiers, Don van Ravenzwaaij.

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
