## [Decision Letter · Decision Letter 0]

6 Jun 2023

PONE-D-23-08280The role of results in deciding to publish

PLOS ONE

Dear Dr. Muradchanian,

Thank you for submitting your manuscript to PLOS ONE. After careful consideration, we feel that it has merit but does not fully meet PLOS ONE’s publication criteria as it currently stands. Therefore, we invite you to submit a revised version of the manuscript that addresses the points raised during the review process.

First of all, I would like to thank all the 3 reviewers. They did a great job in reviewing the paper. I am quite late in sending this decision as one reviewer needed additional time to review the manuscript. I also approved this delay that was fair enough and duly justified. Even if it was quite long for you as an author, I think that it was wise to wait for at least 3 reviewers as all provided very complementary comments on the manuscript. All these comments will help to improve the paper. As the reviewers, I do think that the paper is overall methodologically sounds (this is PLOS criterion for publication), I've chosen to invite you to submit a major revision. Please note that it is clear that the writing needs substantial improvement and that the interpretation need to be tempered and limitations expanded. I really agree with the reviewers on that last point. A very detailed discussion of the limitations is indeed all the more important both in the discussion, but also in the abstract (i.e. a few words about the main limitation). I mean that it is all the more important to avoid any spin (overinterpretation) about your findings. 

We look forward to receiving your revised manuscript.

Kind regards,

Florian Naudet, M.D., M.P.H., Ph.D.

Academic Editor

PLOS ONE

Journal Requirements:

When submitting your revision, we need you to address these additional requirements. 1. Please ensure that your manuscript meets PLOS ONE's style requirements, including those for file naming. The PLOS ONE style templates can be found at https://journals.plos.org/plosone/s/file?id=wjVg/PLOSOne_formatting_sample_main_body.pdf and https://journals.plos.org/plosone/s/file?id=ba62/PLOSOne_formatting_sample_title_authors_affiliations.pdf 2. We note that you have referenced (2.
Cooper H, DeNeve K, Charlton K. Finding the missing science: the fate of studies submitted for review by a human subjects committee. Psychological Methods. 1997; 2; 447-452. https://doi.org/10.1037/1082-989X.2.4.447) which has currently not yet been accepted for publication. Please remove this from your References and amend this to state in the body of your manuscript: (ie “Bewick et al. [Unpublished]”) as detailed online in our guide for authorshttp://journals.plos.org/plosone/s/submission-guidelines#loc-reference-style 

Reviewers' comments:

Reviewer's Responses to Questions

**Comments to the Author**

1. Is the manuscript technically sound, and do the data support the conclusions?

Reviewer #1: Yes

Reviewer #2: Partly

Reviewer #3: Partly

2. Has the statistical analysis been performed appropriately and rigorously? 

Reviewer #1: Yes

Reviewer #2: I Don't Know

Reviewer #3: Yes

3. Have the authors made all data underlying the findings in their manuscript fully available?

Reviewer #1: Yes

Reviewer #2: Yes

Reviewer #3: Yes

4. Is the manuscript presented in an intelligible fashion and written in standard English?

Reviewer #1: Yes

Reviewer #2: Yes

Reviewer #3: No

5. Review Comments to the Author

Reviewer #1: I really like the aims of this manuscript. Although publication bias is well established in biological, medical, social and psychological research, it has not been addressed simultaneously from the perspectives of authros, editors, and reviewers. That said, there are some minor issues and some room for improvement, especially in regards to writing style and structuring.

First, in the abstract the authors should mention the sample size and give at least some statistical results.

Second, the introduction should be shortened a bit and would benefit for being more concise and to the point. Also, the last paragraph of the introduction section is quite odd ("In what follows, ..."). Every academic and reader will know how a scientific paper is structured and what the methods, results, and discussion section are here for. So please remove this paragraph.

Third, in the methods section, the authors should report the final response rate (i.e. participation rate), which unfortunatelly, was quite low.

Fourth, specifications on deviations from the protocol and newly added analyses and their rational was already given in the methods section, therefore these should not be repeated in the results section. These sentences are redundant and should be removed. There are also sentences within the results section that are repetitive and thus should be removed (eg. "The Bayes factors and the 95% credible intervals can be interpreted in the same way as in the corresponding parametric approach" is stated on page 15 and then again on page 17. In addition, the sentence "Because of our doubt on the normality of the difference scores in the three populations (see above), we decided to deviate from our preregistered analysis plan and additionally conduct a non-parametric alternative that relaxes the assumption of normally distributed data: the Bayesian rank-sum test [22]." also repeats itself several times (it is also already mentioned in the methods section, which is sufficient).

Fifth, the following paragraph belongs to the methods section, and not to the results section: "We explored the answers of the respondents to the qualitative optional question regarding why they gave the answers to the quantitative questions about the significant and non-significant results in order to see whether there were specific patterns in the respondents’ reasoning. Based on the answers provided by the respondents to the qualitative question, we made five categories: 1 = p-value is decisive, 2 = p-value is important/relevant but not decisive, 3 = p-value is not important/relevant, 4 = other/unclear (rest category), and 0 = no answer provided to the qualitative question. Authors JM and RH independently classified the answers to the qualitative question into one of these categories. For 45 out of (a total of) 296 answers, there was disagreement on the classification. For each answer for which there was disagreement on classification, authors JM and RH reached consensus in a discussion of no longer than 5 minutes".

Sixth, a major limitation of this study is the very low response rate. This should be mentioned in the limitations section. Social desirability is surely also an issue that should be addressed. I assume that many reviewers and editors who would categorically reject papers with non-significant findings would not honestly admit so in such a survey. The authors should discuss this.

Seventh, just like the introduction (and to some extent also the results section), the manuscript would benefit from a shortened and more concise discussion that is more focused and to the point.

Reviewer #2: The article "Publication Bias in Scientific Literature: Implications and Strategies" tries to disentangle the source of publication bias in the scientific literature at the author, reviewer, and editor level using a survey that provides two short descriptions of scientific papers and the statistical significance of the main effect.

The paper addresses an interesting and important issue; in my opinion, several (methodological) weaknesses should at least be discussed in the paper. As the authors preregistered their analysis plan, which I highly appreciate, I try to keep the suggested post-hoc modifications at a minimum but focus more on discussing potential weaknesses.

In sum, the article would provide more insights into the sources of publication bias, so I recommend major revisions.

1. Clarity and Coherence

The article is mostly written in clear language, however, I suggest to streamline the manuscript further and move some information in the supplemental material. In some parts, the manuscript is a bit lengthy and may profit from a more concise and compact writing style (e.g. the part on the state of research in the introduction).

2. Comprehensive Literature Review

The literature review could be written more concisely, moreover, some studies already tested a similar setting and, therefore should be included in the literature review.

Epstein, W. M. (1990). Confirmational Response Bias Among Social-Work Journals. Science Technology & Human Values, 15(1), 9-38. https://doi.org/10.1177/016224399001500102

Epstein, W. M. (2004). Confirmational Response Bias and the Quality of the Editorial Processes among American Social Work Journals. Research on Social Work Practice, 14(6), 450-458. https://doi.org/10.1177/1049731504265838

3. Methodological Approach

The study's methodology has been preregistered before collecting the data; this is great! The description of the sample was also clearly laid out. In total, the authors approached 1196 journal editors with an editorial request and 3588 journal authors with a reviewer and 3588 journal authors in an author role in a survey.

The survey question evaluated two scientific papers in the abovementioned roles, one with p=.02 and one p=.26. The description in Table 1 contains a lot of duplicate information and may be presented in a more compact manner. The chosen approach has some limitations:

a. The order of the p-values is the same for every respondent (leading to potential order effects that threaten the external validity)

b. Only two p-values are evaluated; this may limit the external validity of the findings. Ideally, also even more minor deviations, e.g. 0.051 vs. 0.049 could be examined. For example in a multifactorial variation of the p-value dimension.

c. The overall response rate of the survey is with 14.3% (171/1196) for editors and 1.7% (60/3588) for reviewers and 1.8% (65/3588) pretty low. This low response rate may induce nonresponse bias, meaning that only naïve authors (that are not aware of the studies goal), which leads to an overestimation of the importance of statistical significance, or most competent authors (that are especially interested in the topic), which lead to an underestimation of the importance of statistical significance take part at the survey.

d. Given your examples, the varied dimension p-value was pretty obvious to the respondents - this could lead to socially desirable answers (potentially underestimating the importance of statistically significant results).

I am aware that the points mentioned above could not be fixed after the data collection. However, the problems that may come with the chosen design should be discussed in more detail.

Regarding the statistical data-analysis a short discussion why Bayesian t-tests were used would be beneficial, especially regarding their advantages compared to frequentist t-tests.

I really value the data and code uploaded by the authors. The analyses conducted in JASP (another statistical software) are, however not documented (is there any code?). From my perspective, changing software during the data-analysis phase should be avoided if possible. As Bayesian inferential methods are implemented in the R package “Bolstad” this could perhaps be avoided in the paper.

4. Minor remarks

• What is the advantage of getting insights into publication bias simultaneously (l. 131 f.) compared to the isolated findings in the literature?

• Although the results comparing reviewers authors and editors are called "ambiguous" l. 39 in the abstract exatly these analysis seem a main goal of the study (cp. L. 135f.).

• "extreme evidence" l. 325 sounds quite strong given the fact that for most researchers it does not seem to matter as described above.

Reviewer #3: Many thanks for the opportunity to review this piece.

The introduction is very long. A lot of what is included there should probably be moved to the Discussion to frame your results in context or removed entirely. The second to last paragraph of your introduction is very redundant and there is no need for the final paragraph of your introduction. I think it can go without saying that we should expect your methods, results, and discussion to follow the introduction. Just combine those two into a simple, brief encapsulation of what your aims .

In the introduction you state that you could only find one study that suggested differential rates of publication based on editorial decisions on the directionality of results (Page 5; Lines 101-105). Song et al. covers a number of studies as part of this review (https://www.journalslibrary.nihr.ac.uk/hta/hta14080/#/abstract) in the section titled “Study Results and Journal Editorial Decisions” but that is also over 10 years old. I assume there has been new research on this since then as well. I would ask the authors to do a more comprehensive search here as that Song review includes in-depth discussion of all the factors of publication bias the authors are interested in. Speaking holistically about the literature makes more sense to me in an introduction rather than cherry-picking individual studies.

My knowledge of Bayesian statistics is limited but from my understanding the statistical methods appear valid.

Your methods for analyzing your qualitative data are in the results section (Page 17) but that all should be in the method section.

Your methods are quite long winded and you repeat elements in the results redundantly. For instance, we don’t need all the hypotheses from the tests stated again in the results. Revising to cut down on extraneous words would also help readability and interpretability. A brief example in which the same information is expressed substantially more concisely:

“The final classification is presented in Table 4. In the second column, it can be observed that 36.1% of the respondents indicated that the p-value is at least important/relevant for publication of study results (counts of the first two rows), whereas 37.8% of the respondents indicated that the p-value is not important/relevant for publication.”

“Of respondents who provided a response on the importance of p-values, 36.1% found them decisive or important for publication, whereas 37.8% did not (Table 4).”

Do you not have any concerns about the very low response rate of your outreach? Per your methods, you invited nearly 1200 editors and 7,200 academics. Your response rate was 14% for editors but just 1.7% for authors and reviewers. You provide no indication as to the characteristics of your sample. Did you at least know what field the responders originated from? Could you have had substantially differential responses between fields? I’m not sure medicine, psychology, and economics are entirely interchangeable in how they view and handle publication bias and this could have a potential impact on your results. While you do indeed find some statistical support for your hypothesis, I think you may be slightly overconfident in your findings given the limitations.

In your Discussion you state that you “found clear evidence for publication bias” (PAge 21, line 446). That is incorrect. You did not text for publication bias directly. You tested for the potential for publication bias based on the opinions of editors, authors, and reviewers.

Overall, I think some of the limitations greatly limit the value of this work, however I do think the findings are interesting even in their limitations and based on my reading it appears methodologically sound. I believe a revision is in order, however, to revise the prose as well as to properly characterize and place these results in context.

6. PLOS authors have the option to publish the peer review history of their article (what does this mean?). If published, this will include your full peer review and any attached files.

Reviewer #1: **Yes: **Michael P. Hengartner

Reviewer #2: **Yes: **Andreas Schneck

Reviewer #3: No

---

## [Author Response · Author response to Decision Letter 0]

8 Jul 2023

Editor 

Thank you for submitting your manuscript to PLOS ONE. After careful consideration, we feel that it has merit but does not fully meet PLOS ONE’s publication criteria as it currently stands. Therefore, we invite you to submit a revised version of the manuscript that addresses the points raised during the review process.

First of all, I would like to thank all the 3 reviewers. They did a great job in reviewing the paper. I am quite late in sending this decision as one reviewer needed additional time to review the manuscript. I also approved this delay that was fair enough and duly justified. Even if it was quite long for you as an author, I think that it was wise to wait for at least 3 reviewers as all provided very complementary comments on the manuscript. All these comments will help to improve the paper. As the reviewers, I do think that the paper is overall methodologically sounds (this is PLOS criterion for publication), I've chosen to invite you to submit a major revision. Please note that it is clear that the writing needs substantial improvement and that the interpretation need to be tempered and limitations expanded. I really agree with the reviewers on that last point. A very detailed discussion of the limitations is indeed all the more important both in the discussion, but also in the abstract (i.e. a few words about the main limitation). I mean that it is all the more important to avoid any spin (overinterpretation) about your findings. 

E-1: We would like to thank you and the reviewers for reading our manuscript and for the comments. We wrote the revised manuscript more concisely, tempered the interpretation, and expanded the limitations. 

Regarding the limitations, we now discuss them in more detail in the discussion section, and added one of the main limitations to our abstract in the following way:

“One of the main limitations was that rather than investigating publication bias directly, we studied potential for publication bias”. 

2. We note that you have referenced (2. Cooper H, DeNeve K, Charlton K. Finding the missing science: the fate of studies submitted for review by a human subjects committee. Psychological Methods. 1997; 2; 447-452. https://doi.org/10.1037/1082-989X.2.4.447) which has currently not yet been accepted for publication. Please remove this from your References and amend this to state in the body of your manuscript: (ie “Bewick et al. [Unpublished]”) as detailed online in our guide for authors

E-2: Our apologies in case we misunderstand this comment, but as far as we know this article was published in Psychological Methods in 1997. 

Reviewer #1

I really like the aims of this manuscript. Although publication bias is well established in biological, medical, social and psychological research, it has not been addressed simultaneously from the perspectives of authros, editors, and reviewers. That said, there are some minor issues and some room for improvement, especially in regards to writing style and structuring.

First, in the abstract the authors should mention the sample size and give at least some statistical results.

R1-1: Thank you very much for reading our manuscript and for your comments. We are happy to hear that you like the aims of our manuscript.

We have now added the sample size and the main statistical results to the abstract in the following way (please see the underlined parts for changes compared to the previous version): 

“Our findings suggest that statistically significant findings have a higher likelihood to be published than statistically non-significant findings, because (1) authors (n = 65) are more likely to write up and submit articles with significant results compared to articles with non-significant results (median effect size 1.10, BF10 = 1.09*107); (2) reviewers (n = 60) give more favourable reviews to articles with significant results compared to articles with non-significant results (median effect size 0.58, BF10 = 4.73*102); and (3) editors (n = 171) are more likely to accept for publication articles with significant results compared to articles with non-significant results (median effect size, 0.94, BF10 = 7.63*107). Evidence on differences in the relative contributions to publication bias by authors, reviewers, and editors is ambiguous (editors vs reviewers: BF10 = 0.31, reviewers vs authors: BF10 = 3.11, and editors vs authors: BF10 = 0.42).”

Second, the introduction should be shortened a bit and would benefit for being more concise and to the point. Also, the last paragraph of the introduction section is quite odd ("In what follows, ..."). Every academic and reader will know how a scientific paper is structured and what the methods, results, and discussion section are here for. So please remove this paragraph.

R1-2: Thank you for pointing this out: we agree that this part could be more concise. In the revised manuscript, we have rewritten and shortened the introduction considerably, and we have removed the last paragraph of the introduction section (see also R2-1 and R3-1). 

Third, in the methods section, the authors should report the final response rate (i.e. participation rate), which unfortunatelly, was quite low.

R1-3: We completely agree. We added the final response rate to the last paragraph in the subsection “Participants” in the methods section in the following way:

“The final response rate was 10.3% for the authors/reviewers and 14.3% for the editors” (see also R2-3-c and R3-5).

Fourth, specifications on deviations from the protocol and newly added analyses and their rational was already given in the methods section, therefore these should not be repeated in the results section. These sentences are redundant and should be removed. There are also sentences within the results section that are repetitive and thus should be removed (eg. "The Bayes factors and the 95% credible intervals can be interpreted in the same way as in the corresponding parametric approach" is stated on page 15 and then again on page 17. In addition, the sentence "Because of our doubt on the normality of the difference scores in the three populations (see above), we decided to deviate from our preregistered analysis plan and additionally conduct a non-parametric alternative that relaxes the assumption of normally distributed data: the Bayesian rank-sum test [22]." also repeats itself several times (it is also already mentioned in the methods section, which is sufficient).

R1-4: After a close rereading, we agree that there was some repetitiveness in the paper. We appreciate the opportunity to clean this up. We have removed duplicate information from the results section (see also R3-4). 

Furthermore, we moved the sentence “The Bayes factors and the 95% credible intervals can be interpreted in the same way as in the corresponding parametric approach” to the method section (subsection “Analyses”), and removed the sentence “Because of our doubt on the normality of the difference scores in the three populations (see above), we decided to deviate from our preregistered analysis plan and additionally conduct a non-parametric alternative that relaxes the assumption of normally distributed data: the Bayesian rank-sum test [22]” from the results section.

Additionally, we shortened the paragraph about respondents’ familiarity with significance testing in the results section (final paragraph), as this information is already provided in the methods section.

Fifth, the following paragraph belongs to the methods section, and not to the results section: "We explored the answers of the respondents to the qualitative optional question regarding why they gave the answers to the quantitative questions about the significant and non-significant results in order to see whether there were specific patterns in the respondents’ reasoning. Based on the answers provided by the respondents to the qualitative question, we made five categories: 1 = p-value is decisive, 2 = p-value is important/relevant but not decisive, 3 = p-value is not important/relevant, 4 = other/unclear (rest category), and 0 = no answer provided to the qualitative question. Authors JM and RH independently classified the answers to the qualitative question into one of these categories. For 45 out of (a total of) 296 answers, there was disagreement on the classification. For each answer for which there was disagreement on classification, authors JM and RH reached consensus in a discussion of no longer than 5 minutes".

R1-5: Agreed. We moved this paragraph to the methods section (subsection “Analyses”; see also R3-3).

Sixth, a major limitation of this study is the very low response rate. This should be mentioned in the limitations section. Social desirability is surely also an issue that should be addressed. I assume that many reviewers and editors who would categorically reject papers with non-significant findings would not honestly admit so in such a survey. The authors should discuss this.

R1-6: Thank you for mentioning these two points. They are now added to the discussion section (the subsection where we discuss the limitations of our study; see also R2-3-c, R2-3-d, and R3-5). 

The limitation regarding the low response rate is added in the following way (please see the underlined part for changes compared to the previous version): 

“There were some limitations related to the surveys used in the present study. The most important one is that some participants mentioned, for instance, via email, that insufficient amount of information (e.g., about the study design, etc.) was provided in the scenarios, or that some relevant information was missing. We had decided to leave out contextual information such that the respondents could only focus on the p-value (assuming that both studies had, for example, a high quality). Additionally, because of this, the purpose of our study was fairly obvious (some respondents let us know by email that they were aware that we were studying publication bias). Perhaps the transparency of the study aim was related to the next limitation of our study, namely the low response rate. We think that there is a possibility that academics might have experienced our survey and the topic of publication bias as uncomfortable, which in turn, resulted in a low response rate.”

The social desirability part is added in the following way: 

“Social desirability might have played a role in the present context. It is possible that some academics systematically refrain from publishing non-significant findings without acknowledging this practice. If this is the case, then our results likely underestimate the reality.” 

Seventh, just like the introduction (and to some extent also the results section), the manuscript would benefit from a shortened and more concise discussion that is more focused and to the point.

R1-7: We agree that the results and the discussion section could be more concise.

In the results section, we have done this as follows: repetitive sentences are removed as much as possible. Answers to the qualitative question are removed from the section where we discuss the more extreme negative differences. Parts in the results section that belong to the methods section are now moved to the methods section (e.g., hypotheses, non-parametric t tests, classification of answers to the qualitative question), including deviations from the preregistered protocol. 

In the discussion section, we have done this in the following way: examples of answers to the qualitative question are removed. The subsection about Registered Reports is shortened. 

However, in the discussion section, certain limitations were missing, so we have now added the following limitations to our discussion section (see also R1-6): 

“Perhaps the transparency of the study aim was related to the next limitation of our study, namely the low response rate. We think that there is a possibility that academics might have experienced our survey and the topic of publication bias as uncomfortable, which in turn, resulted in a low response rate. Furthermore, we evaluated only two p-values, namely p = .02 and p = .26. It would have been interesting to also examine more minor deviations such as p = .049 and p = .051 in order to see what the differences are in the likelihood of endorsing the statistically significant finding for publication compared to the statistically non-significant finding.

Social desirability might have played a role in the present context. It is possible that some academics systematically refrain from publishing non-significant findings without acknowledging this practice. If this is the case, then our results likely underestimate the reality. 

Finally, in the present study, we did not investigate publication bias directly. We studied potential for publication bias based on questionnaires in which we asked what academics would do after reading a simplified fictitious situation. It is important to note that respondents’ answers do not necessarily represent actual behaviour.”

Reviewer #2

The article "Publication Bias in Scientific Literature: Implications and Strategies" tries to disentangle the source of publication bias in the scientific literature at the author, reviewer, and editor level using a survey that provides two short descriptions of scientific papers and the statistical significance of the main effect.

The paper addresses an interesting and important issue; in my opinion, several (methodological) weaknesses should at least be discussed in the paper. As the authors preregistered their analysis plan, which I highly appreciate, I try to keep the suggested post-hoc modifications at a minimum but focus more on discussing potential weaknesses.

In sum, the article would provide more insights into the sources of publication bias, so I recommend major revisions.

R2-0: Thank you very much for critically reading our manuscript. We are happy to hear that you find the discussed issue interesting and important. 

1. Clarity and Coherence

The article is mostly written in clear language, however, I suggest to streamline the manuscript further and move some information in the supplemental material. In some parts, the manuscript is a bit lengthy and may profit from a more concise and compact writing style (e.g. the part on the state of research in the introduction).

R2-1: Thank you for this suggestion. We replaced Table 1 (in the previous version of the manuscript) with the following underlined short description about the scenarios and questions across the three scientific roles in the methods section (subsection “Materials and procedure”):

“The survey started with asking informed consent for all respondents. Then they got presented with a fabricated scenario in which two studies were described, one with a statistically significant result (p = .02) and one with a statistically non-significant result (p = .26). We asked the respondents questions about their intention to write up and submit a paper (author level), recommend publication of a paper (reviewer level), and accept a paper for publication (editor level). Exact formulations of each question are displayed in S1 Table. They were able to provide their answers by using a slider on a scale ranging from 0% to 100%.”

Table 1 from the previous manuscript has now become S1 Table, which is added to the Supporting information. Additionally, the section in the introduction about the state of research is shortened in the revised manuscript. Furthermore, we removed redundant information and reformulated parts to make the manuscript more concise (for specific changes, see R1-2, R1-4, and R1-7). 

2. Comprehensive Literature Review

The literature review could be written more concisely, moreover, some studies already tested a similar setting and, therefore should be included in the literature review.

Epstein, W. M. (1990). Confirmational Response Bias Among Social-Work Journals. Science Technology & Human Values, 15(1), 9-38. https://doi.org/10.1177/016224399001500102

Epstein, W. M. (2004). Confirmational Response Bias and the Quality of the Editorial Processes among American Social Work Journals. Research on Social Work Practice, 14(6), 450-458. https://doi.org/10.1177/1049731504265838

R2-2: We rewrote the literature review more concisely (see also R3-1). Thank you for suggesting the two studies. We think they are relevant, and included them in the literature review.

3. Methodological Approach

The study's methodology has been preregistered before collecting the data; this is great! The description of the sample was also clearly laid out. In total, the authors approached 1196 journal editors with an editorial request and 3588 journal authors with a reviewer and 3588 journal authors in an author role in a survey.

The survey question evaluated two scientific papers in the abovementioned roles, one with p=.02 and one p=.26. The description in Table 1 contains a lot of duplicate information and may be presented in a more compact manner. The chosen approach has some limitations:

R2-3-1: We moved Table 1 to Supporting information as S1 Table. In the method section, we added a short description of the information provided in S1 Table (subsection “Materials and procedure”). Please see also R2-1.

Note that we have added the underlined text to our sample description (subsection "Participants") to avoid ambiguity: “Per editor in our sample (i.e., 210 editors who at least partly filled out our survey in Qualtrics), we randomly selected six authors who published in the journal during the time this editor was editing this journal. In total, after eliminating duplicates, we contacted 1196 editors and 1215 authors/reviewers.”

a. The order of the p-values is the same for every respondent (leading to potential order effects that threaten the external validity)

R2-3-a: The order of the questions about the significant and the non-significant result was actually not the same for every respondent: approximately half of the respondents received the question about the significant result first, and the other half received the question about the non-significant result first (this was done randomly by Qualtrics). We agree that this was not clearly described in the previous version; in order to remedy this, we have added the following underlined sentence to (what is now) S1 Table: 

“The order of the questions about the significant and the non-significant result varied: approximately half of the respondents was presented with the significant result first, and the other half was presented with the question about the non-significant result first.” 

Additionally, we shortly describe this information in the methods section of the revised manuscript (subsection “Materials and procedure”).

b. Only two p-values are evaluated; this may limit the external validity of the findings. Ideally, also even more minor deviations, e.g. 0.051 vs. 0.049 could be examined. For example in a multifactorial variation of the p-value dimension.

R2-3-b: We agree that varying the size of the p-value across a wider range of numbers might have led to additional insights. For the current study, we opted to keep the design relatively simple, so that we could inform respondents that participation would not take long. We have added the idea of additional spread of p-values to the discussion section in the following way: 

“Furthermore, we evaluated only two p-values, namely p = .02 and p = .26. It would have been interesting to also examine more minor deviations such as p = .049 and p = .051 in order to see what the differences are in the likelihood of endorsing the statistically significant finding for publication compared to the statistically non-significant finding.”

c. The overall response rate of the survey is with 14.3% (171/1196) for editors and 1.7% (60/3588) for reviewers and 1.8% (65/3588) pretty low. This low response rate may induce nonresponse bias, meaning that only naïve authors (that are not aware of the studies goal), which leads to an overestimation of the importance of statistical significance, or most competent authors (that are especially interested in the topic), which lead to an underestimation of the importance of statistical significance take part at the survey.

R2-3-c: Please note that the response rate for authors/reviewers was 125/1215=10.3%. Nevertheless, we agree that the response rate is quite low, so we added this limitation to the discussion section (see R1-6). We have also added the final response rate to the method section (subsection “Participants”; see R1-3).

We do not necessarily think that the respondents were naïve, because the aim of the survey was pretty obvious (this was also mentioned by some of the respondents). Additionally, we do not necessarily think that our respondents represent the most interested people in the topic because some people mentioned that significant results are clearly more important than non-significant results. However, we are not certain about our results not being an overestimation or underestimation of the importance of statistical significance. In fact, we do think that social desirability might have played a role in our survey, which can lead to an underestimation. Therefore, we added the following sentences to the discussion section: 

“Social desirability might have played a role in the present context. It is possible that some academics systematically refrain from publishing non-significant findings without acknowledging this practice. If this is the case, then our results likely underestimate the reality” (see also R1-6).

d. Given your examples, the varied dimension p-value was pretty obvious to the respondents - this could lead to socially desirable answers (potentially underestimating the importance of statistically significant results).

R2-3-d: We completely agree. R1 mentioned the same point, please see R1-6 and R2-3-c.

I am aware that the points mentioned above could not be fixed after the data collection. However, the problems that may come with the chosen design should be discussed in more detail.

R2-3-2: Thank you for the above mentioned points. We agree with them and have tried to discuss them as much as possible. 

Regarding the statistical data-analysis a short discussion why Bayesian t-tests were used would be beneficial, especially regarding their advantages compared to frequentist t-tests.

R2-3-3: Thank you very much for mentioning this. We have now added the following part to the methods section (subsection “Analyses”): 

“We decided to conduct Bayesian t tests, as they allow for quantification of evidence in favour of the null hypothesis (relative to a composite alternative hypothesis).”

I really value the data and code uploaded by the authors. The analyses conducted in JASP (another statistical software) are, however not documented (is there any code?). From my perspective, changing software during the data-analysis phase should be avoided if possible. As Bayesian inferential methods are implemented in the R package “Bolstad” this could perhaps be avoided in the paper.

R2-3-4: Unfortunately, it is not possible in JASP to receive the exact code yet. In order to be as transparent as possible, we have submitted the “2023-02-27 Transparency Document.docx” file as part of the category “Other” to PLOS ONE. In this document, the entire data analysis process is described, and it also includes screenshots of the analyses conducted in JASP, so that the reader can reproduce our JASP analyses. This document is also available on OSF: https://osf.io/24d3p

The reason why we had initially decided to conduct our analyses in JASP (and not in R) was because we thought that it might be easier for many readers who are interested in looking into our analyses to understand them better. So we conducted the parametric part in JASP. When we decided to additionally conduct the non-parametric alternatives, the most up to date version of the analytical technique was only available in R. 

4. Minor remarks

• What is the advantage of getting insights into publication bias simultaneously (l. 131 f.) compared to the isolated findings in the literature?

R2-4-1: Thank you for this relevant question. We have now added the following part to the discussion section in which we explain what we think the advantage is of getting insights into publication bias simultaneously compared to the isolated findings in the literature: 

“We think that the advantage of getting insights into publication bias simultaneously is that it enhances the comparability between publication bias induced by authors, by reviewers, and by editors. By doing it simultaneously, it was possible to keep the fictitious scenarios constant across the three roles, and in turn, it was possible to make a direct comparison between publication bias induced by the three scientific roles.”

• Although the results comparing reviewers authors and editors are called "ambiguous" l. 39 in the abstract exatly these analysis seem a main goal of the study (cp. L. 135f.).

R2-4-2: The direct comparison was a main goal of our study, but unfortunately we did not find strong evidence either for a difference between roles or for a lack of difference between roles. We indicate this in the penultimate sentence of the abstract: “Evidence on differences in the relative contributions to publication bias by authors, reviewers, and editors is ambiguous (editors vs reviewers: BF10 = 0.31, reviewers vs authors: BF10 = 3.11, and editors vs authors: BF10 = 0.42).”

• "extreme evidence" l. 325 sounds quite strong given the fact that for most researchers it does not seem to matter as described above.

R2-4-3: The extreme evidence is in favour of treating significant results as more likely to endorse publication than nonsignificant results. The reviewer is correct that there seemed to be a split: for approximately half the respondents, the significance of the results matter, for the other half, it does not. Importantly, for half of the respondents it does matter, and when it does, the pattern invariably favours publication of significant results over publication of nonsignificant results.

Reviewer #3: 

Many thanks for the opportunity to review this piece.

The introduction is very long. A lot of what is included there should probably be moved to the Discussion to frame your results in context or removed entirely. The second to last paragraph of your introduction is very redundant and there is no need for the final paragraph of your introduction. I think it can go without saying that we should expect your methods, results, and discussion to follow the introduction. Just combine those two into a simple, brief encapsulation of what your aims .

R3-1: All three reviewers have made this point, and after rereading the paper, we agree. We wrote our manuscript more concisely and to the point (see also R1-2, R1-4, R1-7, R2-1, R2-2, and R2-3-1).

We combined the second to last paragraph of the introduction section of the previous version of the manuscript and the literature review in the following way (see the underlined parts for changes compared to the previous version): 

“So far, it seems that some direct evidence for publication bias (i.e., directly comparing published and unpublished scientific literature, or following a cohort of scientific studies from their inception [16]) has been provided at both journal (i.e., reviewer and editor) and author level. Additionally, some research has specifically focused on publication bias induced by authors, reviewers, or editors separately [6,11,17,26,27]. Regardless of who has been held responsible for publication bias, it is important to note that it does not occur from a deliberate motive to deceive [3], and academics might not even be aware of these biases [18]. Some have suggested that publication bias may be more pernicious at the author level than at the journal level [10], and some have specifically mentioned that editorial decisions contribute to publication bias [3,16]. However, to the best of our knowledge, this has not been studied directly yet. In the present study, we would like to focus on authors, reviewers, and editors simultaneously in order to see where exactly and to what extent in the process of generating a scientific paper publication bias occurs. Additionally, we would like to make a direct comparison between publication bias induced by authors, reviewers, and editors. By doing so, we will make an attempt to answer the following research question: To what extent do authors, reviewers, and editors contribute to publication bias in scientific literature? We will explicitly focus on whether statistically significant findings have a greater likelihood of being published than statistically non-significant findings.”

In the introduction you state that you could only find one study that suggested differential rates of publication based on editorial decisions on the directionality of results (Page 5; Lines 101-105). Song et al. covers a number of studies as part of this review (https://www.journalslibrary.nihr.ac.uk/hta/hta14080/#/abstract) in the section titled “Study Results and Journal Editorial Decisions” but that is also over 10 years old. I assume there has been new research on this since then as well. I would ask the authors to do a more comprehensive search here as that Song review includes in-depth discussion of all the factors of publication bias the authors are interested in. Speaking holistically about the literature makes more sense to me in an introduction rather than cherry-picking individual studies.

R3-2: The study by Song et al. (2010) is now added to our introduction. Since Song et al. (2010) provide an in-depth discussion of publication bias (and related biases), we decided to add this review to our introduction section. 

We wrote the subsection in the introduction, where we discuss the literature, more concisely. So the part where we initially mentioned that we found one study that focused on editors, is now removed (see also R3-1). Furthermore we added two additional studies (Epstein 1990; 2004) to our introduction section as per Reviewer #2’s suggestion (see R2-2).

My knowledge of Bayesian statistics is limited but from my understanding the statistical methods appear valid.

Your methods for analyzing your qualitative data are in the results section (Page 17) but that all should be in the method section.

R3-3: We agree, so we added this paragraph to the methods section (subsection “Analyses”). See also R1-5.

Your methods are quite long winded and you repeat elements in the results redundantly. For instance, we don’t need all the hypotheses from the tests stated again in the results. Revising to cut down on extraneous words would also help readability and interpretability. A brief example in which the same information is expressed substantially more concisely:

“The final classification is presented in Table 4. In the second column, it can be observed that 36.1% of the respondents indicated that the p-value is at least important/relevant for publication of study results (counts of the first two rows), whereas 37.8% of the respondents indicated that the p-value is not important/relevant for publication.”

“Of respondents who provided a response on the importance of p-values, 36.1% found them decisive or important for publication, whereas 37.8% did not (Table 4).”

R3-4: Thank you for your suggestions. We shortened the methods and the results section (including removing the hypotheses from the results section). See R1-4, R1-7, R2-1, and R2-3-1. 

Furthermore, we replaced the long sentence by the short sentence as suggested. 

Do you not have any concerns about the very low response rate of your outreach? Per your methods, you invited nearly 1200 editors and 7,200 academics. Your response rate was 14% for editors but just 1.7% for authors and reviewers. You provide no indication as to the characteristics of your sample. Did you at least know what field the responders originated from? Could you have had substantially differential responses between fields? I’m not sure medicine, psychology, and economics are entirely interchangeable in how they view and handle publication bias and this could have a potential impact on your results. While you do indeed find some statistical support for your hypothesis, I think you may be slightly overconfident in your findings given the limitations.

R3-5: Reviewer 2 (see R2-3-c) also indicated some confusion about our sampling plan, so we have rewritten the sampling plan in an attempt to remove the ambiguity. Specifically, we have added the following underlined part to our methods section (subsection “Participants”): 

“Per editor in our sample (i.e., 210 editors who at least partly filled out our survey in Qualtrics), we randomly selected six authors who published in the journal during the time this editor was editing this journal. In total, after eliminating duplicates, we contacted 1196 editors and 1215 authors/reviewers.” This means that the response rate for authors/reviewers is 10.3%. The final response rate is now added to the method section (subsection “Participants”), and hopefully this is now clearer to the reader (see R1-3 and R2-3-c). 

We do think that this response rate is still quite low, so we discuss this limitation now in the discussion section (see R1-6 and R2-3-c), but it is, fortunately, not as dramatically low as the 1.7% you mentioned. 

In your Discussion you state that you “found clear evidence for publication bias” (PAge 21, line 446). That is incorrect. You did not text for publication bias directly. You tested for the potential for publication bias based on the opinions of editors, authors, and reviewers.

R3-6: Thank you very much for mentioning this. We removed the corresponding sentence and added the following part to the discussion section: 

“Finally, in the present study, we did not investigate publication bias directly. We studied potential for publication bias based on questionnaires in which we asked what academics would do after reading a simplified fictitious situation. It is important to note that respondents’ answers do not necessarily represent actual behaviour”.

Overall, I think some of the limitations greatly limit the value of this work, however I do think the findings are interesting even in their limitations and based on my reading it appears methodologically sound. I believe a revision is in order, however, to revise the prose as well as to properly characterize and place these results in context.

R3-7: Thank you very much for critically reading our manuscript. We believe that the comments helped us to improve the quality of our manuscript.

---

## [Decision Letter · Decision Letter 1]

8 Aug 2023

PONE-D-23-08280R1The role of results in deciding to publishPLOS ONE

Dear Dr. Muradchanian,

Thank you for submitting your manuscript to PLOS ONE. After careful consideration, we feel that it has merit but does not fully meet PLOS ONE’s publication criteria as it currently stands. Therefore, we invite you to submit a revised version of the manuscript that addresses the points raised during the review process.

First of all, **I would like to thank the 2 reviewers who assessed this new version of the manuscript.** The third reviewer was unavailable but I could check your answers to this reviewer and think that you answered appropriately to this reviewer. **One of the 2 remaining reviewers has remaining questions/requests for clarifications t**hat need to be addressed before I can recommend acceptance. In addition **I have the following requests**: - Please mention in the title the study design ; - Please mention in the abstract the response rate to the survey as an additional limitation ;

- To enhance reporting, please follow an adequate reporting guideline both for the manuscript and the abstract ;

We look forward to receiving your revised manuscript.

Kind regards,

Florian Naudet, M.D., M.P.H., Ph.D.

Academic Editor

PLOS ONE

Reviewers' comments:

Reviewer's Responses to Questions

**Comments to the Author**

1. If the authors have adequately addressed your comments raised in a previous round of review and you feel that this manuscript is now acceptable for publication, you may indicate that here to bypass the “Comments to the Author” section, enter your conflict of interest statement in the “Confidential to Editor” section, and submit your "Accept" recommendation.

Reviewer #1: All comments have been addressed

Reviewer #2: (No Response)

2. Is the manuscript technically sound, and do the data support the conclusions?

Reviewer #1: Yes

Reviewer #2: Partly

3. Has the statistical analysis been performed appropriately and rigorously? 

Reviewer #1: Yes

Reviewer #2: I Don't Know

4. Have the authors made all data underlying the findings in their manuscript fully available?

Reviewer #1: Yes

Reviewer #2: Yes

5. Is the manuscript presented in an intelligible fashion and written in standard English?

Reviewer #1: Yes

Reviewer #2: Yes

6. Review Comments to the Author

Reviewer #1: (No Response)

Reviewer #2: The article "Publication Bias in Scientific Literature: Implications and Strategies" has improved substantively compared to the first version. However, I still have some issues that should be addressed before publication:

Major remarks:

1. The sample composition l. 138 could be reported more clearly, describing the editor and the dependent author/reviewer sample along with the removed duplicates (at best in a table). Please also specify the number of sample persons separately for authors and reviewers (l. 141). The response rate is reported to be 10.3% for both conditions, assuming a half split (607,5) the response rates would be 9,9 or 10,6% (for the authors' sample, there have to be at least 629 sample persons to arrive at a response rate of 10.3%. Please specify the sample sizes and correct the response rate if necessary.

2. Why does Table 4 only contain descriptive results, Fig. 2 in turn, is only mentioned in one sentence and seems to be completely unrelated to the research question

3. I am still not entirely convinced by the use of Bayesian t-tests; what are the major advantages of using this methodology (does it outperform the frequentist t-test in respect of statistical power)? Please elaborate why in l, 207 this specific scale parameter was chosen.

Minor remarks:

l. 161 replace “fabricated scenario” by “hypothetical scenario”

l 186 replace “difference score” by “risk difference” (the effect sizes name)

l, 207 why is exactly this scale parameter chosen?

l. 248 refer to the difference as percentage points

l. 279 and other Tables: Format of tables should be improved (esp. avoiding too many lines)

l. 279 What specific effect size? I suspect Cohen's d?

7. PLOS authors have the option to publish the peer review history of their article (what does this mean?). If published, this will include your full peer review and any attached files.

Reviewer #1: **Yes: **Michael P. Hengartner, PhD

Reviewer #2: **Yes: **Andreas Schneck

---

## [Author Response · Author response to Decision Letter 1]

29 Aug 2023

Editor 

Thank you for submitting your manuscript to PLOS ONE. After careful consideration, we feel that it has merit but does not fully meet PLOS ONE’s publication criteria as it currently stands. Therefore, we invite you to submit a revised version of the manuscript that addresses the points raised during the review process.

First of all, I would like to thank the 2 reviewers who assessed this new version of the manuscript. The third reviewer was unavailable but I could check your answers to this reviewer and think that you answered appropriately to this reviewer. One of the 2 remaining reviewers has remaining questions/requests for clarifications that need to be addressed before I can recommend acceptance. 

In addition I have the following requests: 

- Please mention in the title the study design ; 

- Please mention in the abstract the response rate to the survey as an additional limitation ;

- To enhance reporting, please follow an adequate reporting guideline both for the manuscript and the abstract ;

E-1: We would like to thank you and the reviewers for reading our manuscript and for the comments. 

We have revised the title. The new title is: “The role of results in deciding to publish: A direct comparison across authors, reviewers, and editors based on an online survey”.

In the new abstract, at the end we mention the response rate to the survey as an additional limitation: “Another limitation was the low response rate to the survey.”

To enhance reporting, we made some revisions in our reporting in the abstract and added subheadings to the abstract and manuscript.

Reviewer #2

The article "Publication Bias in Scientific Literature: Implications and Strategies" has improved substantively compared to the first version. However, I still have some issues that should be addressed before publication:

R2-0: Thank you for reading and commenting our manuscript again.

Major remarks:

1. The sample composition l. 138 could be reported more clearly, describing the editor and the dependent author/reviewer sample along with the removed duplicates (at best in a table). Please also specify the number of sample persons separately for authors and reviewers (l. 141). The response rate is reported to be 10.3% for both conditions, assuming a half split (607,5) the response rates would be 9,9 or 10,6% (for the authors' sample, there have to be at least 629 sample persons to arrive at a response rate of 10.3%. Please specify the sample sizes and correct the response rate if necessary.

R2-1: We have collected our data according to a stepwise procedure: 1. We invited 1196 editors, of which 171 responded to the survey. 2. We invited 1215 researchers for the role of reviewer or author, of which 125 responded. 3. Qualtrics randomly assigned each of the respondents in step 2 to the roles of Author (65) and Reviewer (60). We added this information in the following underlined way to the method section (subsection “Participants”), and hopefully it is more clear now: 

“In our analyses, we only included the answers of the respondents who answered the question about the significant result and the question about the non-significant result (see below in the subsection “Materials and procedure” for detailed information about the questions included in our surveys). We collected our samples in a stepwise way: 1) We invited 1196 editors, of which 171 responded to the survey. 2) We invited 1215 researchers for the role of reviewer or author, of which 125 responded. 3) Qualtrics randomly assigned each of the respondents in step 2 to the roles of Author (65) and Reviewer (60). The eventual sample sizes for authors, reviewers, and editors were 65, 60, and 171 respectively. The final response rate was 10.3% for the authors/reviewers and 14.3% for the editors.”

2. Why does Table 4 only contain descriptive results, Fig. 2 in turn, is only mentioned in one sentence and seems to be completely unrelated to the research question

R2-2: We have now added 95% confidence intervals for the average differences in Table 3. Additionally, we have added inferential information (BF10s and 95% credible intervals) when describing the results in Table 3. In order to see how we have performed these analyses (in JASP), see file “2023-08-11 Transparency Document.docx”. In the manuscript (at the end of section “Results”), we have added the following (underlined parts):

“For each category of the qualitative question, we computed the average DS in the likelihood of endorsing publication between the significant and the non-significant results (see Table 3 column 3). We performed exploratory analyses in which we conducted Bayesian independent samples t tests in JASP in order to test the difference in the average DS between category 1 (p-value is decisive) versus category 2 (p-value is important/relevant but not decisive), and between category 2 versus category 3 (p-value is not important/relevant). The null hypotheses for the two t tests stated that there is no difference in the average DS between the two categories, and the alternative hypotheses stated that there is a difference in the average DS between the two categories. For each test we conducted, we calculated Bayes factors that quantify the relative evidence for the alternative hypothesis over the null hypothesis (BF10) provided by the data, and 95% credible intervals based on posterior distributions of the standardized effect size parameters. The results showed that the higher the importance of the p-value indicated in the answer to the qualitative question, the larger the average DS in the likelihood of endorsing publication between the significant and the non-significant results (category 1 versus category 2: BF10 = 5.09*108, 95% credible interval = 1.05, 1.97; category 2 versus category 3: BF10 = 4.02*105, 95% credible interval = 0.54, 1.16). This means that the answers to the qualitative question were in line with the answers to the quantitative questions about the significant and non-significant results. Furthermore, it appeared that the average DS in the categories 4 (i.e., other/unclear [rest category]) and 0 (i.e., no answer provided to the qualitative question) was quite large too.”

Regarding Fig. 2, we think that it is important to include it in our manuscript, because in the preregistration we had mentioned that we would explore to what extent the difference in likelihood of accepting a significant versus a non-significant result was related to familiarity with significance testing. Since the scores on the familiarity with significance testing question appeared to have a ceiling effect, we decided to drop that analysis, and we would like to clarify to the reader why we decided to drop that analysis by showing that figure. However, we do understand that the text in which we mention Fig. 2 is too much separated from the preregistration. Therefore, we replaced the following part from our Results section:

“Finally, the scores on the question about respondents’ familiarity with significance testing are presented in Fig 2. It can be seen that there was not much variation in the answers to this question.”

by:

“As mentioned at the end of the method section, we were planning in our preregistration to explore to what extent the difference in likelihood of accepting a significant versus a non-significant result was related to familiarity with significance testing. We decided to drop this analysis because, as can be observed in Fig. 2, the scores on the familiarity with significance testing question appeared to have a ceiling effect.”

3. I am still not entirely convinced by the use of Bayesian t-tests; what are the major advantages of using this methodology (does it outperform the frequentist t-test in respect of statistical power)? Please elaborate why in l, 207 this specific scale parameter was chosen.

R2-3: Thank you for these comments. Our main argument for using Bayesian t tests (rather than frequentist t tests) is not because the Bayesian t test outperforms the frequentist t test in respect of statistical power. We decided to use Bayesian t tests because they allow for quantification of evidence in favour of the null hypothesis (relative to a composite alternative hypothesis), such as mentioned in subsection “Analyses”. 

We used a Cauchy distribution centred on zero with scale parameter 1/√2, because this is the default prior which is often used in Bayesian null hypothesis testing, and thus makes results comparable across different studies. We agree that this information was missing from the manuscript, and thus we added it to subsection “Analyses” (see the underlined parts):

“The prior distribution for the standardized effect size under the alternative hypothesis was a Cauchy distribution centred on zero with scale parameter 1/√2, which is the default prior that is often used in Bayesian null hypothesis testing, and thus makes results comparable across different studies”.

Minor remarks:

l. 161 replace “fabricated scenario” by “hypothetical scenario”

R2-4: Thank you for this suggestion. We agree that “hypothetical scenario” sounds better, so we replaced “fabricated” by “hypothetical” in the manuscript. 

l 186 replace “difference score” by “risk difference” (the effect sizes name)

R2-5: When writing the manuscript, we struggled with choosing a compatible term for this score. Although we understand that “difference score” might sound a bit vague, we think that “risk difference” may sound a bit negative. Therefore, we decided to make the following adjustment (see underlined parts) to the first paragraph in the subsection “Analyses”:

“... we first computed a difference score (in percentage points) in the likelihood of endorsing publication between the significant and the non-significant results for all respondents (from here on DS).”

l, 207 why is exactly this scale parameter chosen?

R2-6: Please see R2-3.

l. 248 refer to the difference as percentage points

R2-7: We agree with this comment. Please see R2-5.

l. 279 and other Tables: Format of tables should be improved (esp. avoiding too many lines)

R2-8: We have tried to create our tables according to the PLOS ONE formatting requirements as much as possible.

l. 279 What specific effect size? I suspect Cohen's d?

R2-9: Thank you for mentioning this. It should be indeed Cohen’s d. We added this information under Table 1 and Table 2.

---

## [Decision Letter · Decision Letter 2]

18 Sep 2023

The role of results in deciding to publish: A direct comparison across authors, reviewers, and editors based on an online survey

PONE-D-23-08280R2

Dear Dr. Muradchanian,

We’re pleased to inform you that your manuscript has been judged scientifically suitable for publication and will be formally accepted for publication once it meets all outstanding technical requirements.

Kind regards,

Florian Naudet, M.D., M.P.H., Ph.D.

Academic Editor

PLOS ONE

Additional Editor Comments (optional):

Reviewers' comments:

Reviewer's Responses to Questions

**Comments to the Author**

1. If the authors have adequately addressed your comments raised in a previous round of review and you feel that this manuscript is now acceptable for publication, you may indicate that here to bypass the “Comments to the Author” section, enter your conflict of interest statement in the “Confidential to Editor” section, and submit your "Accept" recommendation.

Reviewer #2: All comments have been addressed

2. Is the manuscript technically sound, and do the data support the conclusions?

Reviewer #2: Yes

3. Has the statistical analysis been performed appropriately and rigorously? 

Reviewer #2: Yes

4. Have the authors made all data underlying the findings in their manuscript fully available?

Reviewer #2: Yes

5. Is the manuscript presented in an intelligible fashion and written in standard English?

Reviewer #2: Yes

6. Review Comments to the Author

Reviewer #2: Dear all,

many thanks for adressing all my concerns and for reading your interesting and relevant paper!

All the best

Andreas Schneck

7. PLOS authors have the option to publish the peer review history of their article (what does this mean?). If published, this will include your full peer review and any attached files.

Reviewer #2: **Yes: **Andreas Schneck

---

## [Editor Report · Acceptance letter]

25 Sep 2023

PONE-D-23-08280R2 

The role of results in deciding to publish: A direct comparison across authors, reviewers, and editors based on an online survey 

Dear Dr. Muradchanian:

I'm pleased to inform you that your manuscript has been deemed suitable for publication in PLOS ONE. Congratulations! Your manuscript is now with our production department. 

Kind regards, 

on behalf of

Pr. Florian Naudet 

Academic Editor

PLOS ONE